# SAM-GUIDED UNSUPERVISED DOMAIN ADAPTATION FOR 3D SEGMENTATION

## ABSTRACT

Unsupervised domain adaptation (UDA) in 3D segmentation tasks presents a formidable challenge, primarily stemming from the sparse and unordered nature of point cloud data. Especially for LiDAR point clouds, the domain discrepancy becomes obvious across varying capture scenes, fluctuating weather conditions, and the diverse array of LiDAR devices in use. While previous UDA methodologies have often sought to mitigate this gap by aligning features between source and target domains, this approach falls short when applied to 3D segmentation due to the substantial domain variations. Inspired by the remarkable generalization capabilities exhibited by the vision foundation model, SAM, in the realm of image segmentation, our approach leverages the wealth of general knowledge embedded within SAM to unify feature representations across diverse 3D domains, and further solves the 3D domain adaptation problem. Specifically, we harness the corresponding images associated with point clouds to facilitate knowledge transfer and propose an innovative hybrid feature augmentation methodology, which significantly enhances the alignment between the 3D feature space and SAM's feature space, operating at both the scene and instance levels. Our method is evaluated on many widely-recognized datasets, and achieves state-of-the-art performance.

## 1 INTRODUCTION

3D scene understanding is fundamental for many real-world applications, such as autonomous driving, robotics, smart cities, etc. Based on the point cloud, 3D segmentation is a critical task for scene understanding, which requires assigning semantic labels for each point. Current deep learning-based solutions (Zhu et al., 2021; Xu et al., 2023) rely heavily on massive annotated data, which are high-cost and lack generalization capability for handling domain shifts. Unsupervised domain adaptation is significant for alleviating data dependency. However, unlike images with dense and regular representation, point clouds, especially LiDAR point clouds of large scenes, are unstructured and sparse, and have overt differences in patterns for various capture devices. Although some studies (Yi et al., 2021; Saltori et al., 2022; Shaban et al., 2023) have extended 2D techniques to solve the 3D UDA problem, the performance is still limited due to the essential defect of point cloud representation.

Considering that RGB cameras yield dense, color-rich, and structured data, and more importantly, they represent minor discrepancies across various devices, certain 3D UDA methods (Jaritz et al., 2020; Cardace et al., 2023; Cao et al., 2023) utilize the synergy of LiDAR and camera capabilities to achieve more comprehensive and precise perception, and further enhance adaptation capabilities for 3D segmentation tasks. However, these methods usually train 2D and 3D networks simultaneously, which are difficult to converge and demand substantial online computing resources.

Vision foundation models (VFMs), such as the Segment Anything Model (SAM) (Kirillov et al., 2023), have garnered significant attention due to their remarkable performance in addressing open-world vision tasks. Such models are trained on massive image data with tremendous parameters. Compared with a common model trained on limited data, VFMs have more general knowledge and much stronger generalization capability. Many works such as Chen et al. (2023b;a) have emerged recently to transfer the general 2D vision knowledge of VFMs to 3D and have achieved promising performance.

Based on SAM, focusing on image segmentation, we propose a novel paradigm for 3D UDA segmentation. As shown in Fig. 1, different from previous UDA approaches that strive to align the target

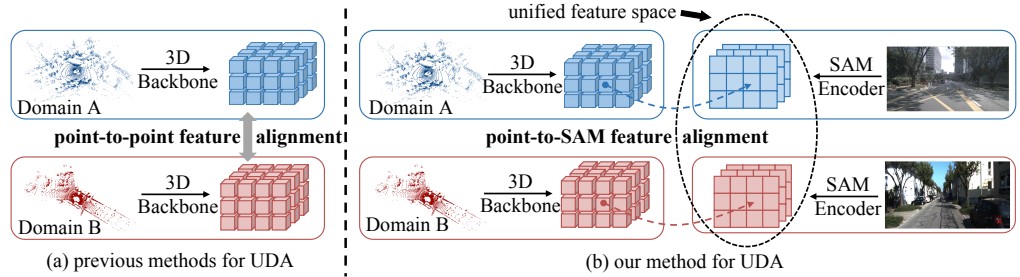

Figure 1: Comparison of 3D UDA paradigms. Different from aligning two feature domains directly, our method makes both the source domain and target domain align with the SAM feature space.

domain to the source domain so that the model trained on labeled source data can also work on target data without annotation, our method makes both the source domain and target domain align with the SAM feature space. SAM feature space contains more general knowledge, which provides a friendly space to unify the feature representation from different domains. We utilize RGB images to assist point clouds in our framework. However, unlike methods mentioned above only using images to provide auxiliary information, we take images as a bridge to align diverse 3D feature spaces to the SAM feature space, so we do not need to train extra 2D networks and can process the image offline for less computing resources. Considering that the 3D feature space created by the source-domain data and the target-domain data is still much smaller than the SAM feature space, we propose a hybrid feature augmentation method at both scene and instance levels to generate more 3D data with diverse feature patterns in a broader data domain, which can further benefit the 3D-to-SAM feature alignment. In particular, we make full use of the masks generated by SAM to mix instance-level point clouds with the other domains. This technique can maintain the geometric completeness of instances, which is beneficial for semantic recognition.

To verify the effectiveness of our method, we compare it with current SOTA works on extensive 3D UDA segmentation settings and our method outperforms others by a large margin, improving about $14\%$ mIoU for VirtualKITTI-to-SematicKITTI, about $15\%$ mIoU for Waymo-to-nuScenes, and about $20\%$ mIoU for nuScenes-to-SemanticKITTI domain adaptation. Surprisingly, our unsupervised method achieves comparable performance with the supervised method for city-changing and light-changing settings on the nuScenes dataset. Furthermore, we also test our method on more challenging tasks, such as panoptic segmentation and domain generalization, and experimental results show that our method is robust and has good generalization capability.

In summary, our contributions are as follows:

- We propose a novel unsupervised domain adaptation approach for 3D segmentation, leveraging the foundational model SAM to guide the alignment of features from diverse 3D data domains into a unified domain.
- We introduce a hybrid feature augmentation strategy at both scene and instance levels, generating more distinct feature patterns across a broader data domain for better feature alignment.
- We conduct extensive experiments on large-scale datasets and achieve SOTA performance.

## 2 RELATED WORK

### 2.1 POINT CLOUD SEMANTIC SEGMENTATION

Point cloud semantic segmentation (Zhu et al., 2021; Guo et al., 2020) is a rapidly evolving field, and numerous research works have contributed to advancements in this area. The pioneering approach PointNet (Qi et al., 2017a) directly processes point clouds without voxelization and revolutionizes 3D segmentation by providing a novel perspective on point cloud analysis. Further, PointNet++ (Qi et al., 2017b) extends PointNet with hierarchical feature learning through partitioning point clouds into local regions. To handle sparse point cloud data efficiently within large-scale scenes, a framework called SparseConvNet (Graham et al., 2018) has been specifically crafted. It excels in processing sparse 3D data and has been effectively utilized in various applications, including 3D semantic

segmentation. MinkUNet (Choy et al., 2019) represents a significant advancement in point cloud semantic segmentation. Employing multi-scale interaction networks, MinkUNet enhances the segmentation of point clouds, effectively addressing the challenges posed by 3D spatial data. Our 3D segmentation networks are the popular SparseConvNet and MinkUNet. Due to the sparse characteristics of point cloud data, many current methods (Yan et al., 2022; Krispel et al., 2020; He et al., 2022) add corresponding dense image information to facilitate point cloud segmentation tasks. Our method also uses image features to assist point cloud segmentation, and additionally, we take advantage of the 2D segmentation foundation model to achieve effective knowledge transfer.

## 2.2 DOMAIN ADAPTATION FOR 3D SEGMENTATION

Unsupervised Domain Adaptation (UDA) aims at transferring knowledge learned from a source annotated domain to a target unlabelled domain, and there are already several UDA methods proposed for 2D segmentation (Chang et al., 2019; Zhang & Wang, 2020; Kim & Byun, 2020; Zou et al., 2018). In recent years, domain adaptation techniques have gained increasing traction in the context of 3D segmentation tasks. Yi et al. (2021) leverage a "Complete and Label" strategy to enhance semantic segmentation of LiDAR point clouds by recovering underlying surfaces and facilitating the transfer of semantic labels across varying LiDAR sensor domains. CosMix (Saltori et al., 2022) introduces a sample mixing approach for UDA in 3D segmentation, which stands as the pioneering UDA approach utilizing sample mixing to alleviate domain shift. It generates two new intermediate domains of composite point clouds through a novel mixing strategy applied at the input level, mitigating domain discrepancies. However, due to the sparsity and irregularity of the point cloud, the disparity across different point cloud data domains is larger compared to that across 2D image domains, which makes it difficult to mitigate the variation across domains.

With the development of multi-modal perception (Bai et al., 2022; Cong et al., 2023) in autonomous driving, prevalent 3D datasets (Fong et al., 2022; Mei et al., 2022; Behley et al., 2019; Geyer et al., 2020) include both 3D point clouds and corresponding 2D images, making leveraging multi-modality for addressing domain shift challenges in point clouds convenient. xMUDA (Jaritz et al., 2020; 2022) shows the power of combining 2D and 3D networks within a single framework, which achieves outstanding performance by aggregating the scores from these two branches. This achievement is attributed to the complementary nature resulting from the diverse modalities processed by each branch. Peng et al. (2021) introduce Dynamic Sparse-to-Dense Cross-Modal Learning (DsCML) to enhance the interaction of multi-modality information, ultimately boosting domain adaptation sufficiency, while Cardace et al. (2023) elucidate this complementarity of image and point cloud through an intuitive explanation centered on the effective receptive field, and proposes to feed both modalities to both branches. However, in practice, training two networks with distinct architectures is difficult to converge and demands substantial computing resources due to increased memory. Our method uses the pre-trained foundation model to process the image data, guaranteeing the quality of the image features and enabling the training process to focus on the 3D model.

## 2.3 VISION FOUNDATION MODELS

The rise of foundation models (Devlin et al., 2018; Jia et al., 2021; Touvron et al., 2023) has garnered significant attention which are trained on extensive datasets, consequently demonstrating exceptional performance. Foundation models (Zou et al., 2023; Wang et al., 2023b) have seen significant advancements in the realm of 2D vision, and several research studies have been conducted to leverage this progress and extend these foundation models to comprehend 3D information. Representative works CLIP (Radford et al., 2021) leverage contrastive learning techniques to train both text and image encoders, CLIP2Scene (Chen et al., 2023b) extends the capabilities of CLIP by incorporating a 2D-3D calibration matrix, facilitating a deeper comprehension of 3D scenes. The Meta Research team recently launched the 'Segment Anything Model' (Kirillov et al., 2023), trained on an extensive dataset of over 1 billion masks from 11 million images. Utilizing efficient prompting, SAM can generate high-quality masks for image instance segmentation. The integration of flexible prompting and ambiguity awareness enables SAM with robust generalization capabilities for various downstream segmentation challenges. Many methods (Chen et al., 2023a;b; Liu et al., 2023) take it as an off-the-shelf tool and distillate the knowledge to solve 3D problems by 2D-3D feature alignment. In our work of tackling the UDA of 3D segmentation, we utilize SAM to provide 2D prior knowledge for 3D feature alignment in a wider data domain.

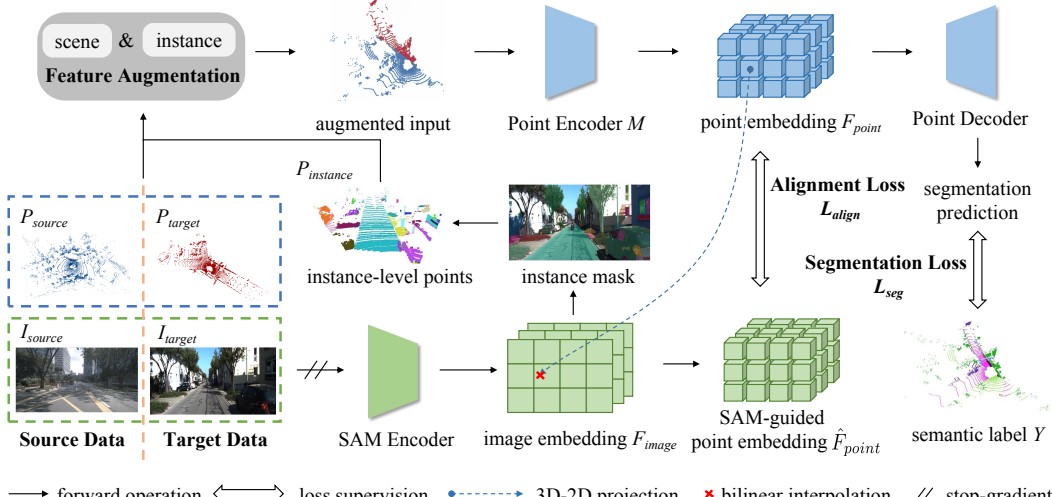

Figure 2: Pipeline of our method. The point cloud is fed into the point encoder for point embeddings at the top, and the corresponding images are passed through the SAM encoder for image embeddings at the bottom, from which we obtain SAM-guided point embedding with the 2D-3D projection. Alignment loss is calculated from the SAM-guided features and original features. Furthermore, augmented inputs by mixing source and target data, providing diverse feature patterns, are utilized to boost the 3D-to-SAM feature alignment.

## 3 METHOD

### 3.1 PROBLEM STATEMENT

We explore UDA for 3D segmentation, in which we have the source domain, denoted as $D_S = \{P_S, I_S, Y_S\}$ with paired input, namely point cloud $P_S$ and image $I_S$, as well as annotated labels $Y_S$ for each point, and the target domain denoted as $D_T = \{P_T, I_T\}$ without any annotation. Using these data, we train a 3D segmentation model that can generalize well to the target domain.

3D data from different domains have obvious differences in distribution and patterns, leading to over-fitting problems when models trained in one domain try to analyze data from another. The main challenge in 3D UDA is to extract useful features despite these domain differences, essentially aligning features between these distinct domains. Our solution is to map data from different domains into a unified feature space, ensuring the model performs consistently across domains.

### 3.2 FRAMEWORK OVERVIEW

The visual foundation model SAM is trained by massive image data, which contains relative general vision knowledge and provides a friendly feature space to unify diverse feature representations. Taking 2D images as the bridge, the 3D feature space of different domains can be indirectly unified by bringing them closer to the SAM feature space based on 2D-to-3D knowledge transfer. Based on this, we design a novel SAM-guided UDA method for 3D segmentation, as shown in Fig. 2.

Specifically, given a point cloud input $P$, the point encoder $M$ generates a point embedding $F_{point} \in \mathbb{R}^{n \times d}$ in the $d$-dimensional latent feature space. Concurrently, the corresponding image input $I$ is passed through the SAM encoder for a c-channels image embedding $F_{image} \in \mathbb{R}^{h \times w \times c}$. Utilizing the correspondence between the point cloud and image, we acquire SAM-guided point embedding $\hat{F}_{point} \in \mathbb{R}^{n \times d}$ to compute the alignment loss $L_{align}$ with the original point embedding $F_{point}$, serving the purpose of using SAM as a bridge to integrate the features of diverse data domains into a unified feature space. Notably, during training, the input for feature alignment consists of data from both source and target domains. We named this process as **SAM-guided Feature Alignment**. At the same time, as for labeled data $Y$, segmentation loss $L_{seg}$ is also calculated as semantic supervision. During model training, only the point cloud branch of the whole pipeline is trained, and the gradient is not calculated in the image branch, which makes our method more lightweight. Furthermore, the

point mix-up strategy trains a network on additional data derived from the convex combination of the source domain and target domain, thus effectively reducing domain shifts in UDA. These additional data can also provide augmented features of the intermediate domain between the source and target domains for SAM-guided feature alignment. Beyond the simple concatenation of two point clouds at the scene level, we use the instance mask output by SAM to select instances' point clouds from two domains for mixing up so that local instance-level geometric features can be better maintained to project to the semantic space. This strategy is called **Hybrid Feature Augmentation at Scene and Instance level**. During inference, only point cloud data $P$ is needed for prediction from the 3D network. Details of our method are introduced below.

### 3.3 SAM-GUIDED 3D FEATURE ALIGNMENT

Previous UDA methods usually align the feature space of the target domain to that of the source domain so that the model trained on the source domain with labeled data can also recognize the data from the novel domain. However, the distributions and patterns of 3D point clouds in various datasets have substantial differences, making the alignment very difficult. SAM (Kirillov et al., 2023), a 2D foundation model, is trained with a huge dataset of 11M images, granting it robust generalization capabilities to address downstream segmentation challenges effectively. If we can align features extracted from various data domains into the unified feature space represented by SAM, the model trained on the source domain can effectively handle the target data with the assistance of the universal vision knowledge existing in the SAM feature space.

We focus on training a point-based 3D segmentation model, while SAM is a foundation model trained on 2D images, which presents a fundamental challenge: how to bridge the semantic information captured in 2D images with the features extracted from 3D points. Most outdoor large-scale datasets with point clouds and images provide calibration information to map the 3D points into the corresponding images. With the projection matrix $R_{L2C}$, we can easily translate the coordination of points $P$ from the 3D LiDAR coordinate system to the 2D image coordinate system. This transformation can be formally expressed as

$$P_{image} = R_{L2I}P_{lidar}.$$

Once we calculate the 2D positions of points in the image coordinate system, we can determine their corresponding positions in the SAM-guided image embedding $F_{image}$, which is generated from the image by the SAM feature extractor. As the positions of points in the image embedding typically are not integer values, we perform bilinear interpolation based on the surrounding semantic features in the image embedding corresponding to the point, which to some extent alleviates the effect of calibration errors and allows us to derive the SAM-guided feature of each point, denoted as

$$\hat{F}_{point} = \textbf{Bilinear}(F_{image},\ P_{image}).$$

Then, the original point embedding $F_{point}$ from both source and target domains are all required to align with their corresponding SAM-guided features $\hat{F}_{point}$. Specifically, we utilize the cosine function to measure the similarity of $F_{point}$ and $\hat{F}_{point}$, employing it as the alignment loss $L_{align}$ during training. With the supervision of $L_{align}$, features obtained by the point encoder $M$ will gradually converge towards the feature space represented by SAM, achieving the purpose of extracting features within a unified feature space from the input of different domains. The formulation of the loss function for feature alignment is shown below:

$$L_{align} = 1 - \textbf{cos}(\hat{F}_{point},\ F_{point}).$$

### 3.4 HYBRID FEATURE AUGMENTATION AT SCENE AND INSTANCE LEVEL

3D point features of the source-domain data and the target-domain data only cover subsets of the 3D feature space, which are limited to align with the whole SAM feature space with more universal knowledge. Therefore, more 3D data with diverse feature patterns in a broader data domain is needed to achieve more effective 3D-to-SAM feature alignment.

Previous works (Xiao et al., 2022; Kong et al., 2023) usually focus on synthesizing data by combining data in the source domain and target domain at the scene level, including polar-based, range-based, and laser-based, as shown in Fig.3(a). Polar-based point mix-up selects semi-circular point

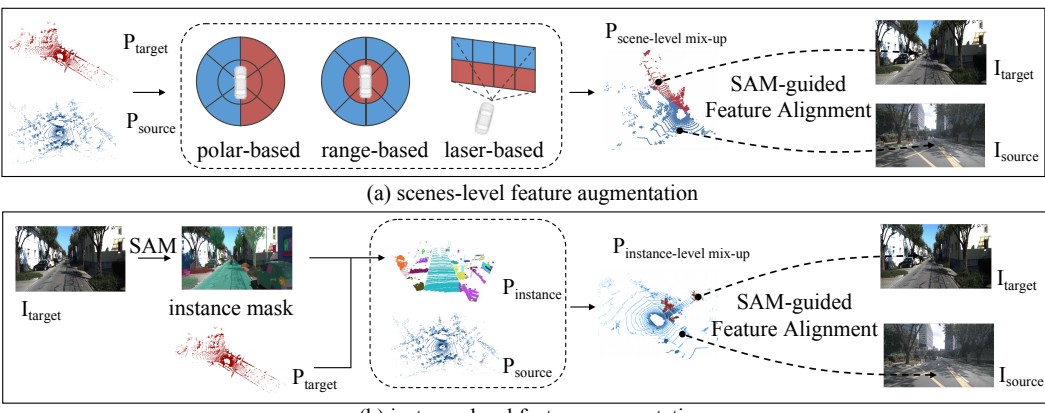

Figure 3: Hybrid feature augmentation by data mixing for better 3D-to-SAM feature alignment.

cloud data from two different domains based on the polar coordinates of the point cloud. Range-based point mix-up divides the point cloud by its distance from the center, synthesizing circular point data close to the center and ring point data farther away from the center. Laser-based point mix-up determines the part of point clouds based on the number of laser beams, combining points with positive and negative laser pitch angles from different domains for synthesis. These ways of scene-level feature augmentation can maintain the general pattern of LiDAR point clouds as much as possible and improve the data diversity. Moreover, they are simple to process without any requirement for additional annotations such as real or pseudo-semantic labels. We adopt these three kinds of scene-level data augmentation in our method.

However, scene-level data augmentation will, to some extent, destroy the completeness of the point cloud of instances in the stitching areas and affect the exploitation of local geometric characteristics of point clouds. To further increase the data diversity and meanwhile keep the instance feature patterns of LiDAR point clouds for better semantic recognition, we propose an instance-level augmentation method. Benefiting from the instance mask output from SAM, we can thoroughly exploit the instance-level geometric features. Compared with pre-trained 3D segmentation models, SAM provides more accurate and robust instance masks and enables us to avoid extra warm-up for a pre-train model, simplifying the whole training process. Therefore, we perform instance-level data synthesis as Fig.3(b) shows. Specifically, we begin by employing SAM to generate instance masks for input images from either the source or target domain (We take target data as the example in the figure). Next, we use the calibration matrix to project the corresponding point cloud into the image. The instance information of each point is determined according to whether the projection position of the point cloud falls within a specific instance mask, and then we randomly select some points with $20 \sim 30$ specific instances, mixed with the point cloud from the other domain by direct concatenation to achieve point augmentation at the instance level.

In practice, we combine all the ways of feature augmentation at both scene level and instance level with a random-selection strategy for a more comprehensive feature augmentation, which generates a more diverse set of point cloud data with varied feature patterns. Then, the augmented points are fed into the point encoder $M$ to obtain the point embedding $F_{point}$ with distinct feature patterns in a broader data domain beyond the source domain and target domain for more effective SAM-guided feature alignment. Notably, to maintain the consistency of the point cloud and the image, we extract SAM-guided point embedding based on the original image embedding corresponding to the point. The formulation of feature augmentation is shown below:

$$F_{point} = M(\mathbf{Aug}(P_{source}, P_{target}))$$

## 4 EXPERIMENT

We first introduce datasets and implementation details. After that, we explore several domain shift scenarios and compare them with current SOTA UDA methods for 3D segmentation. Then, we conduct extensive ablation studies to give a comprehensive assessment of the submodules of our method. Finally, we extend our method to more challenging tasks to show its generalization capability.

## 4.1 DATASET SETUP

We first follow the benchmark introduced in xMUDA (Jaritz et al., 2022) to evaluate our method, comprehending four domain shift scenarios, including (1) USA-to-Singapore, (2) Day-to-Night, (3) VirtualKITTI-to-SemanticKITTI and (4) A2D2-to-SemanticKITTI. The first two leverage nuScenes (Caesar et al., 2020) as their dataset, consisting of 1000 driving scenes in total with 40k annotated point-wise frames. Specifically, The former differs in the layout and infrastructure while the latter exhibits severe illumination changes between the source and the target domain. The third is more challenging since it is the adaptation from synthetic to real data, implemented by adapting from VirtualKITTI (Gaidon et al., 2016) to SemanticKITTI (Behley et al., 2019) while the fourth involves A2D2 (Geyer et al., 2020) and SemanticKITTI as different data domains, where the domain discrepancy lies in the distinct density and arrangement of 3D point clouds captured by different devices since the A2D2 is captured by 16-beam LiDAR and the SemanticKITTI uses 64-beam LiDAR. For the above settings, noted that only 3D points visible from the camera are used for training and testing, specifically, only one image and corresponding points for each sample are used for training.

Since we only use the image combined with SAM as assistance for the training of a 3D segmentation network instead of training a new 2D segmentation network, we focus on comparing the performance of the 3D segmentation network and enabling model training with the whole point cloud sample even if some part of it is not visible in the images. Thus, we also compare our method with others trained with the whole 360° view of the point cloud, in which three datasets are involved including nuScenes, SemanticKITTI, and Waymo (Mei et al., 2022). In these settings, we use 6 images in nuScenes covering 360° view, 1 image in SemanticKITTI covering 120° view, and 5 images in Waymo covering 252° view. More information is in Sec. A. For metric, We compute the Intersection over the Union (IoU) for each class and report the mean Intersection over the Union (mIoU).

## 4.2 IMPLEMENTATION DETAILS

We make source and target labels compatible across these experiments. For all benchmarks in prior multi-modal UDA methods, we strictly follow class mapping like xMUDA for a fair comparison, while we map the labels of the dataset in other experiments into 10 segmentation classes in common. Our method is implemented by using the public PyTorch (Paszke et al., 2019) repository MMDetection3D (Contributors, 2020) and all the models are trained on a single 24GB GeForce RTX 3090 GPU. To compare fairly, we use SparseConvNet (Graham et al., 2018) with U-Net architecture as the 3D backbone network when comparing with all the multi-modal methods followed by xMUDA and use MinkUNet32 (Choy et al., 2019) when compare with the state-of-the-art uni-modal method CosMix. For the image branch, the ViT-h variant SAM model is utilized to generate image embedding for SAM-guided feature alignment and instance masks for hybrid feature augmentation in an offline manner. We keep the proportion of mixed data and normal data from the source and target domain the same during model training. Before the data is fed into the 3D network, data augmentation such as vertical axis flipping, random scaling, and random 3D rotations are widely used like all the compared methods. For the model training strategies, we choose a batch size of 8 for both source data and target data, then mix the data batch for training at each iteration. Besides, we adopt AdamW as the model optimizer and One Cycle Policy as the learning-rate scheduler.

## 4.3 EXPERIMENTAL RESULTS AND COMPARISON

Tab. 1 and Tab. 2 show the experimental results and performance comparison with previous UDA methods for 3D segmentation under the setup introduced in Sec. 4.1. Each experiment contains two reference methods in common, a baseline model named **Source only** trained only on the source domain and an upper-bound model named **Oracle** trained only on the target data with annotations. Tab. 1 focuses on four domain shift scenarios introduced by Jaritz et al. (2022) and comparison with these multi-modal methods based on xMUDA such as Peng et al. (2021) and Cardace et al. (2023). Among them, MM2D3D fully exploits the complementarity of image and point cloud and proposes to feed two modalities to both branches, achieving better performance. Our method outperforms it by +6.8% (USA → Singapore), +0.3% (Day → Night), +14.6% (v.KITTI → Sem.KITTI), +6.0% (A2D2 → Sem.KITTI) respectively, because our method aligns all the features into a unified feature space with the guidance of SAM instead of simply aligning features from image and point cloud in 2D and 3D network. Tab. 2 focuses on the scenarios where not all the point clouds are visible

Table 1: Results under four domain shift scenarios introduced by xMUDA. We report all the 3D network performance of compared multi-modal UDA methods in terms of mIoU.

| Method | USA → Singapore | | Day → Night | | v.KITTI → Sem.KITTI | | A2D2 → Sem.KITTI | |
|---|---|---|---|---|---|---|---|---|
| Source only | 62.8 | +0.0 | 68.8 | +0.0 | 42.0 | +0.0 | 35.9 | +0.0 |
| xMUDA (Jaritz et al., 2022) | 63.2 | +0.4 | 69.2 | +0.4 | 46.7 | +4.7 | 46.0 | +10.1 |
| DsCML (Peng et al., 2021) | 52.3 | −10.5 | 61.4 | −7.4 | 32.8 | −9.2 | 32.6 | −3.3 |
| MM2D3D (Cardace et al., 2023) | 66.8 | +4.0 | 70.2 | +1.4 | 50.3 | +8.3 | 46.1 | +10.2 |
| **Ours** | **73.6** | **+10.8** | **70.5** | **+1.7** | **64.9** | **+22.9** | **52.1** | **+16.2** |
| Oracle | 76.0 | − | 69.2 | − | 78.4 | − | 71.9 | − |

Table 2: Results under four domain shift scenarios with 360° point cloud, where not all the points are visible in the images. We report the 3D network performance in terms of mIoU.

| Method | nuScenes → Sem.KITTI | | Sem.KITTI → nuScenes | | nuScenes → Waymo | | Waymo → nuScenes | |
|---|---|---|---|---|---|---|---|---|
| Source only | 27.7 | +0.0 | 28.1 | +0.0 | 29.4 | +0.0 | 21.8 | +0.0 |
| PL (Morerio et al., 2017) | 30.0 | +2.3 | 29.0 | +0.9 | 31.9 | +2.5 | 22.3 | +0.5 |
| CoSMix (Saltori et al., 2023) | 30.6 | +2.9 | 29.7 | +1.6 | 31.5 | +2.1 | 30.0 | +8.2 |
| MM2D3D (Cardace et al., 2023) | 30.4 | +2.7 | 31.9 | +3.8 | 31.3 | +1.9 | 33.5 | +11.7 |
| **Ours** | **48.5** | **+20.8** | **42.9** | **+14.8** | **44.9** | **+15.5** | **48.2** | **+26.4** |
| Oracle | 70.3 | − | 78.3 | − | 79.9 | − | 78.3 | − |

in the images and we re-implement three methods by their official codes. Morerio et al. (2017) uses the prediction from the pre-trained model as pseudo labels for unlabelled data to retrain this model, which is widely used in UDA methods. Saltori et al. (2023) trains a 3D network with only the utilization of a point cloud, which generates new intermediate domains through a mixing scene-level strategy to mitigate domain discrepancies. MM2D3d is the SOTA multi-modal method as described above, but it needs all the points visible in the image for the best performance. Our method surpasses them by at least +17.9% (nuScenes → Sem.KITTI), +11.0% (Sem.KITTI → nuScenes), +13.0% (nuScenes → Waymo), +14.7% (Waymo → nuScenes) respectively by a large margin, since hybrid feature augmentation can provide more intermediate domains and SAM-guided feature alignment can help map the whole point cloud into the unified feature space. For qualitative comparison shown in Fig. 4, predictions in the ellipses demonstrate that source-only and MM2D3D models often infer wrong and mingling results, especially for the person category, while our method can provide correct and more fine-grained segmentation. More qualitative results are in Sec. C

## 4.4 ABLATION STUDY

To show the effectiveness of each module of our method, we conduct ablation studies on nuScenes-to-SemanticKITTI UDA. We also show the effect of other visual foundation models on our method.

Table 3: Ablation study. Baseline means the result of the source-only model indicating the lower-bound and Pseudo Label means re-training the model with pseudo labels.

| Setting | Baseline | SAM-guided Feature Alignment | Hybrid Feature Augmentation | | Pseudo Label | mIoU |
|---|---|---|---|---|---|---|
| | | | Scene-level | Instance-level | | |
| (1) | ✓ | | | | | 27.7 |
| (2) | ✓ | ✓ | | | | 34.0 |
| (3) | ✓ | | ✓ | ✓ | | 28.6 |
| (4) | ✓ | ✓ | ✓ | | | 40.1 |
| (5) | ✓ | ✓ | | ✓ | | 39.0 |
| (6) | ✓ | ✓ | ✓ | ✓ | | 44.0 |
| (7) | ✓ | ✓ | ✓ | ✓ | ✓ | **48.5** |

**Effectiveness of Model Components** We first analyze the effects of all the submodules in our method in Tab. 3, containing SAM-guided Feature Alignment, Hybrid Feature Augmentation, and Pseudo Label. SAM-guided Feature Alignment aligns all the point features with the corresponding feature embeddings output by SAM, guiding the 3D network map point cloud into the unified feature space represented by SAM while Hybrid Feature Augmentation generates additional point cloud data of the intermediate domain for feature extraction to maximize the effect of feature alignment. Setting (1), (2), (3), and (6) in the table shows that combining the two submodules improves performance by a large margin. Besides, re-training the model with pseudo labels is a strategy widely used in UDA tasks and it also improves the performance.

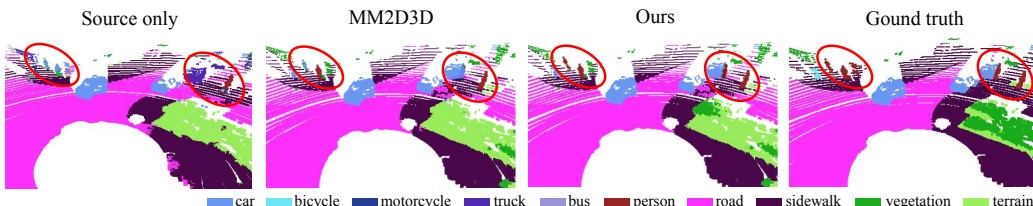

Figure 4: Visualization results of the domain adaptation from nuScenes to SemanticKITTI.

Table 5: Extension on more challenging tasks, such as UDA for Panoptic Segmentation(left) and Domain Generalization(right), where N, S, A represent nuScenes, SemanticKITTI and A2D2 dataset.

| Task | Method | PQ | PQ$^{\dagger}$ | RQ | SQ | mIoU |
|---|---|---|---|---|---|---|
| nuScenes → Sem.KITTI | Source only | 14.0 | 21.6 | 19.9 | 55.8 | 27.7 |
| | PL | 15.9 | 22.7 | 22.2 | 58.1 | 29.7 |
| | **Ours** | **34.3** | **38.4** | **42.6** | **55.9** | **48.5** |
| | Oracle | 50.5 | 52.2 | 57.8 | 77.2 | 70.3 |
| Sem.KITTI → nuScenes | Source only | 15.6 | 22.1 | 20.7 | 52.7 | 28.2 |
| | PL | 16.8 | 23.0 | 21.7 | 48.3 | 29.0 |
| | **Ours** | **24.6** | **30.7** | **30.8** | **60.0** | **42.9** |
| | Oracle | 40.7 | 44.9 | 47.2 | 83.8 | 78.3 |

| Method | N,S → A |
|---|---|
| Baseline | 45.0 |
| xMUDA (Jaritz et al., 2022) | 44.9 |
| Dual-Cross (Li et al., 2022) | 41.3 |
| BEV-DG (Li et al., 2023) | 55.1 |
| **Ours** | **57.2** |

**Effectiveness of Hybrid Feature Augmentation** As for the detailed ablation of feature alignment, we adopt hybrid strategies for diverse data with distinct feature patterns, which not only mix up points at the scene level in polar-based, range-based, and laser-based ways but also at the instance level with the help of instance mask output by SAM. Random selection in all these point mix-up ways forms this feature augmentation. Setting (4), (5), (6) in Tab. 3 shows that both mix-up methods can help feature alignment with more distinct features but the hybrid strategy raises the best performance. Further ablations of the Hybrid Feature Augmentation are in Sec. B.

**Effectiveness of Visual Foundation Model** One of the insightful designs in our method is that we leverage the visual foundation model to provide a potential unified feature space indirectly aligning point feature. Except for SAM, there exist other visual foundation models such as InternImage (Wang et al., 2023a) serving for image-based

Table 4: Generalize with other VFMs.

| baseline | InternImage | SAM |
|---|---|---|
| 27.7 | 36.9 | **44.0** |

tasks like classification and segmentation. We replace the SAM's image encoder with InternImage's to guide the feature alignment in the same way. Tab. 4 shows that the performance still can be improved compared with the baseline, but not as effective as SAM because SAM is a segmentation-specific foundation model trained with much more data and contains more general knowledge.

## 4.5 MORE CHALLENGING TASKS

Since we achieve the purpose of mapping data from different domains into a unified feature space, the extracted feature can be used for some more challenging tasks. We show some extension results of our method in Tab. 5. The left subtable shows the results of UDA for panoptic segmentation, a more challenging task requiring instance-level predictions. With more accurate and fine-grained semantic prediction, our method achieves promising results. The right subtable shows the results of domain generalization, in which target data only can be used for testing. With the ability of stronger data-to-feature mapping, our method outperforms the current SOTA method (Li et al., 2023). In the future, we seek to explore the potential of our method on more tasks, such as 3D detection.

## 5 CONCLUSION

In this paper, we acknowledge the limitations of existing UDA methods in handling the domain discrepancy present in 3D point cloud data and propose a novel paradigm to unify feature representations across diverse 3D domains by leveraging the powerful generalization capabilities of the vision foundation model, SAM, significantly enhancing the adaptability of 3D segmentation models. Hybrid feature augmentation strategy is also proposed to use the instance semantics of SAM for better 3D-SAM feature alignment. We conduct extensive experiments under several UDA scenarios, showing that our method surpasses all the compared state-of-the-art methods by a large margin.

## ETHICS STATEMENT

3D segmentation has been a long-standing research area in computer vision. Our work in unsupervised domain adaptation for 3D segmentation contributes positively to the community by alleviating the cost of extensive annotation. The large-scale dataset used in our study are released by other researchers and has been publicly available for years. The dataset primarily focuses on data related to autonomous driving scenarios, including road, cars, pedestrians, and more. To the best of our knowledge, this dataset does not contain any privacy-related information.

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

APPENDIX

In this section, We provide more supplementary material to support the findings and observations drawn in the main body of this paper.

## A    DATASET DETAILS

We conduct extensive experiments under several domain shift scenarios with five well-known large-scale datasets, including nuScenes (Caesar et al., 2020), SemanticKITTI (Behley et al., 2019), VirtualKITTI (Gaidon et al., 2016), A2D2 (Geyer et al., 2020), and Waymo (Mei et al., 2022). All of these datasets provide point clouds and corresponding images captured by distinct devices resulting in different data representations, thus we will give more detailed information about them. (1) nuScenes contains 1000 driving scenes with 20 seconds for each scene, taken at 2Hz. The scenes are split into train (28,130 keyframes), validation (6,019 keyframes), and corresponding point-wise 3D semantic labels provided by nuScenes-Lidarseg. (2) SemanticKITTI features a large-angle front camera and a 64-layer LiDAR and the captured data from scenes 0, 1, 2, 3, 4, 5, 6, 7, 9, and 10 are used for training while scenes 8 as validation at most experiments. Notably, in the experiment of Tab. 1, we use the split introduced by xMUDA (Jaritz et al., 2022) for fair comparison. (3) VirtualKITTI consists of 5 driving scenes created with the Unity game engine. VirtualKITTI does not simulate LiDAR but rather provides a dense depth map with semantic labels, so we use the 2D-to-3D projecting to generate a point cloud from the depth map. (4) A2D2 consists of 20 drives corresponding to 28,637 frames. The point cloud comes from three 16-layer front LiDARs (left, center, right), and the semantic labeling was carried out in the 2D image. (5) Waymo offers 2,860 temporal sequences captured by five cameras and one LiDAR in three different geographical locations, leading to a total of 100k labeled data, making it larger than existing datasets that offer point-wise segmentation labels. We visualize these five datasets with one sample to show the data domain difference in Fig. 5.

Table 6: Effect of different point mix-up strategies for the hybrid feature augmentation. "No Mix-up" applies SAM-guided feature alignment with no-mixed data from the source and target domain.

| Setting | No Mix-up | Scene-level Feature Augmentation | | | Instance-level Feature Augmentation | mIoU |
|---|---|---|---|---|---|---|
| | | Polar-based | Range-based | Laser-based | | |
| (1) | ✓ | | | | | 34.0 |
| (2) | | ✓ | | | | 37.0 |
| (3) | | | ✓ | | | 38.5 |
| (4) | | | | ✓ | | 37.6 |
| (5) | | | | | ✓ | 40.1 |
| (6) | | ✓ | ✓ | ✓ | | 39.0 |
| (7) | | ✓ | ✓ | ✓ | ✓ | **44.0** |

## B    MORE EXPERIMENTAL DATA

We conduct further ablation on the same setting in Sec.4.4 for hybrid feature augmentation to thoroughly analyze its effect. We take all the point mix-up strategies used in it apart to conduct separate experiments and Setting (2), (3), (4), (5) in Tab. 6 shows that all of them contribute to our method. So we combine them with random selection to form the hybrid feature augmentation. Setting (2), (3), (4), (5) in Tab. 6 show this comprehensive approach boosts the performance since it can provide different data with distinct feature patterns as much as possible for SAM-guided feature alignment. Additionally, we found that this approach can be used in other multi-modal methods as a normal data augmentation method. Taking MM2D3D (Cardace et al., 2023) as an example, we add this module into it and Tab. 7 shows that our feature augmentation approach also improves its performance. However, this feature augmentation approach contributes to our method in a better way because of our effective SAM-guided feature alignment.

Table 7: Generalization of feature augmentation for other multi-modal UDA methods.

| Source only | MM2D3D (Cardace et al., 2023) | MM2D3D w/ feature augmentation | Ours w/o feature augmentation | Ours |
|---|---|---|---|---|
| 27.7 | 30.4 | 33.2 | 34.0 | 48.5 |

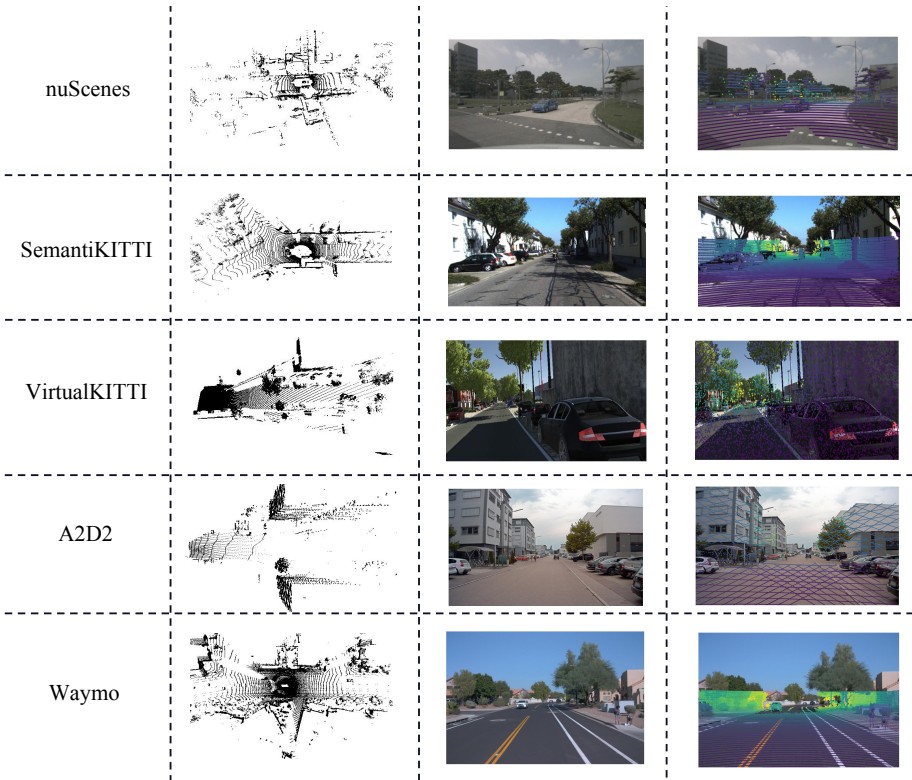

Figure 5: Visualization of each used dataset. From left to right, the figure shows the point cloud, one image corresponding to the point cloud, and the projection of the point cloud on the image.

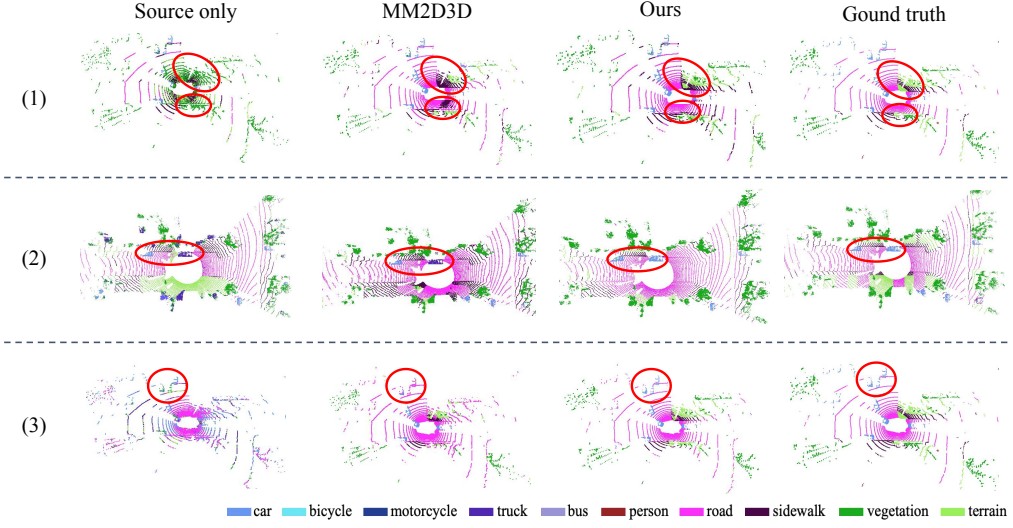

Figure 6: Visualization results of the domain adaptation from more domain shift scenarios, including (1)SemanticKITII-to-nuScenes, (2) nuScenes-to-Waymo, and (3) Waymo-to-nuScenes.

## C    MORE QUALITATIVE RESULTS

Except for the visualization results of the domain adaptation from nuScenes to SemanticKITII shown in the main paper in Fig. 4, we provide additional qualitative results in Fig. 6 representing other introduced domain shift scenarios in the experiment including SemanticKITII-to-nuScenes, nuScenes-to-Waymo, and Waymo-to-nuScenes. Our method has more accurate predictions for the cars, sidewalk, terrain, etc.

