# OpenReview forum: "SAM-guided Unsupervised Domain Adaptation for 3D Segmentation"
_ICLR.cc/2024/Conference — ICLR 2024 Conference Withdrawn Submission_

### Official Review · Reviewer_Zd6y · 2023-10-26

**Soundness:** 3 good
**Presentation:** 3 good
**Contribution:** 2 fair
**Rating:** 3
**Confidence:** 4

**Summary:**

A domain adaptation method is proposed by the author, which leverages the generalization capabilities of the large-scale model SAM. The method. The author proposes a feature enhancement technique that operates on both scene-wide and individual elements. The conducted experiments show the effectiveness of the method.

**Strengths:**

This paper exhibits a comprehensive structure and clear expression.

The authors have demonstrated through experimentation that their method achieves excellent results.

**Weaknesses:**

The feature alignment and alignment proposed in the paper are common solutions in this task, lacking in innovation.

The paper lacks insight into the field, and some points motivated in the introduction section are not validated by experiments. For instance, "these methods usually train 2D and 3D networks simultaneously, which are difficult to converge and demand substantial online computing resources..." The authors should provide in-depth analysis and visualized results to substantiate their motivation.

**Questions:**

see weakness

---

### Official Review · Reviewer_SoFF · 2023-10-31

**Soundness:** 3 good
**Presentation:** 3 good
**Contribution:** 2 fair
**Rating:** 6
**Confidence:** 3

**Summary:**

Authors focus on point cloud (pcl) classification, especially in a different domain than training, enhanced by the “features” from the image segmentation model. Point cloud features are extracted from pcl and image features are extracted from the corresponding image using the SAM.

Datasets are with known transformation between 3d pcl and their corresponding 2d images. Domain adaptation is performed in the SAM feature space extracted from the image and can be “linked” to 3D pcl.

Method is evaluated on LIDAR’s KITTI dataset.

**Strengths:**

I like the idea where 3D data are aligned in the features space using the image segmentation model. Moreover the fact that alignment and SAM features are needed only during the training (and not in interference)  seems to be very efficient.  It looks like such an application to UDA Point cloud segmentation is novel.

Method is clearly explained.

**Weaknesses:**

I believe the 3.1. Problem Statement is not precise. In addition to D_S and D_T, authors use knowledge extracted from 11M images (Sec 3.3) - SAM. Are comparisons fair?

The process that the loss is not propagated to SAM and thus the SAM is not updated looks to me like a weak point. It has a feeling that authors didn’t do it (I’m not sure weather they have access to do it) and then they try to sell it as a strength.

**Questions:**

My biggest concern is about the fairness of the evaluation. Authors use the SAM dataset (knowledge) but competitors do not. If this problem stays I would be towards rejection (as it will require big changes), if it will be shown that it is ok then I will lean towards accept.

---

### Official Review · Reviewer_z4wG · 2023-11-04

**Soundness:** 3 good
**Presentation:** 2 fair
**Contribution:** 2 fair
**Rating:** 5
**Confidence:** 5

**Summary:**

This paper investigates the unsupervised domain adaptation problem in 3D semantic segmentation, which aims to mitigate the distribution shift between two closely-related datasets.
The core contribution of this work is two-fold:
-  Taking the SAM-encoded images features as the proxy to align two domains within an unified feature space.
- In addition to the existing mixup/cosmix paradigm (scene level), the authors employ the SAM-derived instance maskes to devise the instance-level mixup paradigm.

**Strengths:**

1, This paper is generally well-organized, and motivations of core designs are well presented.
2, Extensive experiments are conducted to verify the effectiveness of the proposed method.

**Weaknesses:**

1, The core contribution of this paper, SAM for domain alignment in 3D segmentation and instance+scene mixup augmentation is limited in the novety.
- The camera to lidar projecction is widely used in 3D segmentation/detection [a,b,c], and the proposed SAM-guided Feature Alignment is simply to impose the alignment loss between the 2d and 3d features. And according to Table 3, the performance gain mainly benefit from the foundation model, +6.7, while the  proposed augmentation paradigm can bring less, i.e., + 0.9.

[a] Liu et al. BEVFusion: Multi-Task Multi-Sensor Fusion with Unified Bird's-Eye View Representation, ICRA 2023.

[b] Liang et al. BEVFusion: A Simple and Robust LiDAR-Camera Fusion Framework.  NeuIPS 2022.

[c] Philion et al. Lift, Splat, Shoot: Encoding Images From Arbitrary Camera Rigs by Implicitly Unprojecting to 3D. ECCV 2022.


- The proposed instance-level mixup/cosmix strategy is also been developped in 3d detection works, which is known as the copy-paste augmentation.
    Zhang et al. Multi-Modality Cut and Paste for 3D Object Detection. ICCV19 .


2, Missing critical references
 - 1 The projection between 3d point cloude space and 2D images shoudl be more explicitly explained, including the fomula of projection matrix, along with proper references. For example, the utilizatiion of intrinsic/extrinsic parameter of the cameres.
 - 2 one of the adaptation methods for 3D segmentation are missing, which at least should be discussed.

        Li et al. 	Adversarially Masking Synthetic to Mimic Real: Adaptive Noise Injection for Point Cloud Segmentation Adaptation.

**Questions:**

1, Are all reported results with retraining on pseudo labels? This can leads to unfair comparison since not all methods adopts the two stage schemes.

2, In table 3, it would be interesting to see the comparion on instance-level vs. scene level augmentions without the SAM's involvement, i.e., (4) (5) without SAM--guided augmentation.

3, As you explained in Sec. 4.1, only one dataset have camera images covers 360 degreess. How to you perrform the alingment in the uncovered regions? This should be  briefly explained.

4, Conituning with Q3, despite not that necessary, I think it would be interesting to investigate the impact of cover rate (degress of the camera covers) regarding the SAM-based alignment, which can help us better capture how the knowledge of SAM is transferred.

---

### Official Review · Reviewer_qceh · 2023-11-06

**Soundness:** 2 fair
**Presentation:** 2 fair
**Contribution:** 2 fair
**Rating:** 6
**Confidence:** 3

**Summary:**

This paper presents a method for Unsupervised Domain Adaptation (UDA) for semantic segmentation on LIDAR point clouds. It uses a pre-trained foundation model, SAM to align point clouds from different domains. SAM is used to segment images associated with point clouds, and the image segment LIDAR points are projected to, are used to get the SAM feature for the particular point. A separate Point-encoder to encode point clouds is used to compare the SAM feature against. Point cloud features are made to align closely with the SAM features and doing this for both the domains in turn brings the point cloud features close to each other, with the assumption that SAM features are similar across domains. Additionally, scene and instance level point mix-up is used to combine training data from both domains. Results obtained exceed SoTA for UDA for a number of LIDAR semantic segmentation datasets.

**Strengths:**

This work uses a sound concept and assumption: that SAM features generalize well across datasets and use their association with LIDAR points to bring point cloud features across datasets closer to each other.
The paper is reasonably well written, with a detailed literature review.
Results obtained exceed SoTA for UDA on semantic segmentation.

**Weaknesses:**

The paper lacks detail about the network structures and exactly how the features are combined.
The equations are not labelled/numbered.

**Questions:**

I have a number of questions about Figure 2.
There is a dotted line between the image embedding F_image and the point embedding F_image to show 3D-2D projection. But it is not clear how the images and point clouds from source and target domains are combined, what is done at the seam, etc. There is a red cross in the image embedding showing bilinear interpolation; presumably this means that when points from the point cloud are projected into the image, bilinear interpolation is used.
But the arrow is between the point and image embeddings. So are we projecting the point embedding to the image embedding, or are we projecting the 3D LIDAR point into the image (which is what I presume we would do in the case of 3D->2D projection)
Table 1: it is somewhat suspicious that the proposed method exceeds the Oracle (where labels are available for the target domain) for the Day->Night and vKITTI->Semantic KITTI tasks.
The P_image = R_L2I P_lidar equation is a gross simplification of what actually happens, which involves rotating and translating the LIDAR coordinate frame into the camera coordinate frame and then projecting the points with the camera intrinsics.

---

### Official Review · Reviewer_yHAr · 2023-11-08

**Soundness:** 4 excellent
**Presentation:** 4 excellent
**Contribution:** 2 fair
**Rating:** 5
**Confidence:** 4

**Summary:**

The authors propose an approach leveraging the powerful vision foundation model, SAM, to perform UDA in 3D segmentation. By harnessing SAM's generative capabilities, the proposed method can align distributions between the source and target domains better. To further enrich the data, the authors propose instance-level feature augmentation to alleviate the weakness of the scene-level augmentation. The results are significantly improved over state-of-the-arts on various datasets.

**Strengths:**

1. The paper is clear and easy to follow.
2. Applying SAM in the 3D domain in a clean way.
3. Elegant way to utilze SAM for instance-level segmentation to alleviate the weakness of the common methods.
3. There is a significant performance gain on various datasets.

**Weaknesses:**

The novelty of the work is limited. The benefits seem to primarily stem from the use of SAM. The authors should either conduct additional experiments to show that the performance gain does not solely result from utilizing SAM or provide rigorous analyses explaining why SAM is crucial in this work.

**Questions:**

1. Table 3 indicates that SAM is the primary factor contributing to the method's strong performance. Is it possible to apply SAM to previous state-of-the-art methods? I believe that the authors should conduct experiments in a more equitable manner by either applying SAM to previous state-of-the-art methods or running the proposed method with limited computational budgets similar to those used in previous works.
2. As I mentioned in question 1, the authors should provide the computation cost for each method for table 1 and 2.
3.  **Effectiveness of Visual Foundation Model** at Sec. 4.4 should be expanded.
    - What will the performance be if we use other CNN-based segmantation models for the image branch?
    - Table 1 and 2 should also include the proposed method with different backbones.
4. In Sec 3.3, the authors mentioned that aligning 3D point clouds across datasets is challenging. It would be great if the authors could provide some insights and rigorous analyses on why SAM can solve the issue, besiedes raw performance gain.